# Involvement of Mitochondrial Dysfunction, Endoplasmic Reticulum Stress, and the PI3K/AKT/mTOR Pathway in Nobiletin-Induced Apoptosis of Human Bladder Cancer Cells

**DOI:** 10.3390/molecules24162881

**Published:** 2019-08-08

**Authors:** Yih-Gang Goan, Wen-Tung Wu, Chih-I Liu, Choo-Aun Neoh, Yu-Jen Wu

**Affiliations:** 1Department of Surgery, Kaohsiung Veterans General Hospital Pingtung Branch, Pingtung 91202, Taiwan; 2Division of Thoracic Surgery, Department of Surgery, Kaohsiung Veterans General Hospital, Kaohsiung 81362, Taiwan; 3Department of Nursing, Meiho University, Pingtung 91202, Taiwan; 4Department of Food Science and Nutrition, Meiho University, Pingtung 91202, Taiwan; 5Department of Research, Pingtung Christian Hospital, Pingtung 90059, Taiwan; 6Department of Biological Technology, Meiho University, Pingtung 91202, Taiwan; 7Yu Jun Biotechnology Co., Ltd., Kaohsiung 81363, Taiwan

**Keywords:** nobiletin, mitochondrial dysfunction, endoplasmic reticulum stress, PI3K/AKT/mTOR pathway

## Abstract

Nobiletin (NOB) is a polymethoxylated flavonoid isolated from citrus fruit peel that has been shown to possess anti-tumor, antithrombotic, antifungal, anti-inflammatory and anti-atherosclerotic activities. The main purpose of this study was to explore the potential of using NOB to induce apoptosis in human bladder cancer cells and study the underlying mechanism. Using an MTT assay, agarose gel electrophoresis, a wound-healing assay, flow cytometry, and western blot analysis, this study investigated the signaling pathways involved in NOB-induced apoptosis in BFTC human bladder cancer cells. Our results showed that NOB at concentrations of 60, 80, and 100 μM inhibited cell growth by 42%, 62%, and 80%, respectively. Cells treated with 60 μM NOB demonstrated increased DNA fragmentation, and flow cytometry analysis confirmed that the treatment caused late apoptotic cell death. Western blot analysis showed that mitochondrial dysfunction occurred in NOB-treated BFTC cells, leading to cytochrome *C* release into cytosol, activation of pro-apoptotic proteins (caspase-3, caspase-9, Bad, and Bax), and inhibition of anti-apoptotic proteins (Mcl-1, Bcl-xl, and Bcl-2). NOB-induced apoptosis was also mediated by regulating endoplasmic reticulum stress via the PERK/elF2α/ATF4/CHOP pathway, and downregulating the PI3K/AKT/mTOR pathway. Our results suggested that the cytotoxic and apoptotic effects of NOB on bladder cancer cells are associated with endoplasmic reticulum stress and mitochondrial dysfunction.

## 1. Introduction

Urothelial carcinoma of the transitional epithelium is also called transitional cell carcinoma (TCC), and can develop in the urothelium of the renal pelvis, ureter, or bladder. Urothelial carcinoma is the fifth most malignant tumor in the world, and the second most common cancer of the genitourinary tract [1,2]. Clinical evidence has demonstrated that urothelial carcinoma accounts for approximately 90–95% of all bladder cancers, and therefore more than 90% of human bladder tumors are TCCs [3,4]. TCC is an important disease in the aging population, especially as the proportion of women suffering from cancer increases with age [5]. The routes of TCC metastasis to nearby tissues include the local invasion of perivesical tissues and lymphatic spread to other sites [6]. The most common treatment for bladder TCC is surgical removal, followed by immunotherapy and chemotherapy. Bacillus Calmette-Guérin (BCG) is the most commonly-used agent for bladder perfusion therapy. When BCG is used in immunotherapy for the treatment of bladder cancer, it stimulates the host immune system, which is thought to be the mechanism by which the tumor is destroyed [7,8]. In recent years, radiation therapy and concurrent systemic chemotherapy have been proven to be effective in patients with bladder cancer for whom radical cystectomy is unsuitable [9,10]. However, every treatment method has its limitations, and the development of new diagnostic techniques and treatments for TCC remains an urgent issue.

Chemotherapy is a treatment in which certain drugs are employed to directly kill cancer cells or to stop them from growing and spreading, and its mechanism involves causing apoptosis. Chemotherapy is currently the most conducive to the development of new cancer treatments [11]. The two major apoptotic responses in mammalian cells are the intrinsic and extrinsic pathways, and signaling of the intrinsic pathway occurs in the mitochondria and endoplasmic reticulum (ER) [12,13]. During apoptosis, pro-apoptotic Bcl-2 family proteins, such as Bax, change the mitochondrial membrane potential and elicit release of apoptotic factors from mitochondria. Cytochrome *c* is one of the key factors released from the outer surface of the inner mitochondrial membrane and is subsequently released into the cytoplasm during apoptosis. Once in the cytosol, cytochrome *C* further activates caspase-9, which then leads to activation of downstream caspase-3. The active caspases cleave cellular protein poly(ADP-ribose) polymerase-1 (PARP-1) to destroy the apoptotic cells [14,15]. The PI3K/AKT/mTOR signaling pathway plays an important role in apoptosis, cell proliferation, differentiation, and survival. When PI3K is activated, it triggers the activations of a series of AKT downstream proteins and mTOR, which initiates the expressions of critical regulatory genes through regulating the transcription of p70 [16,17].

Nobiletin (NOB), a flavonoid found in tangerines, is a polymethoxylated flavonoid that has been shown to possess anti-tumor, antithrombotic, antifungal, anti-inflammatory and anti-atherosclerotic activities [18,19,20,21,22]. NOB also has a neurotrophic action, and has been demonstrated to improve memory impairment and pathology in a mouse model of Alzheimer′s disease [23,24,25,26]. NOB has a weak anti-proliferative activity in normal cell lines, but possesses a strong activity to inhibit the proliferation of several cancer cell lines [27]. NOB reduces the tumor-invasive activity of human fibrosarcoma HT-1080 cells through suppressing the expressions of matrix metalloproteinase-1 (MMP-1) and MMP-9 [28], and exerts inhibitory effects on the production of MMP-1, -3 and -9 in rabbit synovial fibroblasts in vitro [29]. In a mouse model, NOB prevents peritoneal dissemination of human gastric carcinoma in SCID mice [30]. These findings suggested that NOB has the potential to be developed as a new natural anti-tumor drug. In this study, we aimed to investigate the effect and mechanism of NOB in human bladder cancer cells.

## 2. Results

### 2.1. Effect of Nobiletin (NOB) on the Growth of BFTC Bladder Cancer Cells

Using an MTT assay, the cytotoxic effect of NOB at various concentrations (20, 40, 60, 80, and 100 μM) on BFTC bladder cancer cells were examined. The results showed that at concentrations ranging from 60 to 100 μM, BFTC cell growth was significantly inhibited, and the inhibitory effect was positively correlated with the NOB concentration (Figure 1A). NOB at concentrations of 60, 80, and 100 μM had a cell growth inhibitory effect of 42%, 62%, and 80%, respectively. In this concentration range, the higher NOB concentration, the greater the inhibition of BFTC cell growth. In this study, we used different concentrations of NOB (20, 40, and 60 μM) in the remaining experiments.

The apoptotic effect of NOB on BFTC cells was assessed via electrophoretic DNA analysis using agarose gel. The results showed that the degree of DNA fragmentation also increased with the concentration-dependent manners of NOB (Figure 1B), suggesting that the DNA damage that induces apoptosis is correlated with the concentration of NOB. NOB exhibited a growth delay effect on BFTC cells. As shown in Figure 1C, compared with the control cells, treatment with 20, 40, and 60 μM NOB caused decreased cell colony numbers, by 24%, 58%, and 71%, respectively. The results indicated that an increased concentration of NOB had a greater inhibition effect on cell proliferation of BFTC cells. We also employed a wound-healing assay to examine the effects of NOB on cell proliferation. The results showed that in cultures treated with higher concentrations of NOB, wound closure was slower (Figure 1D), demonstrating that NOB suppressed BFTC cell proliferation.

### 2.2. NOB Induced Apoptosis in BFTC Bladder Cancer Cells

In order to confirm that the cytotoxic effect of NOB on BFTC cells is exerted through the induction of apoptosis, we used fluorescein isothiocyanate (FITC) Annexin-V and propidium iodide (PI) labelling with flow cytometry to analyze cell apoptosis. The results showed that in control cells and cells treated with NOB at 20, 40, and 60 μM, early apoptotic cells accounted for 0.91%, 1.22%, 1.13%, and 2.22%, respectively; these values did not differ significantly (Figure 2). However, late apoptotic cells accounted for 5.15%, 6.46%, 8.78%, and 10.5%, respectively, showing that the proportion of late apoptotic cells was positively correlated with the concentration of NOB. The results demonstrated that NOB did not cause early apoptosis, but induced late apoptosis in BFTC cells.

### 2.3. NOB-Induced Apoptosis Was Associated with Mitochondria Inactivation in BFTC Cells

Apoptosis can be initiated by either the extrinsic pathway or the intrinsic pathway in cells. The intrinsic pathway is triggered by stress on cellular organelles, especially mitochondria and the endoplasmic reticulum (ER). Mitochondria are considered the key control centers of apoptosis, and are known to regulate apoptotic pathway-related proteins, including caspase-3, caspase-9, PARP-1, cytochrome *C*, Bax, Bad, *p*-Bad, Bcl-2, Bcl-xl, and Mcl-1. In order to study the mechanism of NOB-induced apoptosis in BFTC bladder cancer cells, we used western blot analysis to investigate the changes in the levels of these proteins after NOB treatment. The results showed that under treatment with higher concentrations of NOB, the expressions of Bax, Bad, and cytochrome *C* were increased, while the expressions of *p*-Bad, Bcl-xl, Bcl-2, and Mcl-1 were decreased (Figure 3). The pattern of change of these proteins confirmed that NOB-induced apoptosis in BFTC cells was correlated with Bcl-2 family proteins. It could be inferred that Bax (pro-apoptotic) and Bcl-2 (anti-apoptotic) regulation of homeostasis in cells plays a key role in the release of cytochrome *C* from the mitochondria to the cytoplasm, which further activates downstream caspase activities.

In order to investigate whether NOB-induced apoptosis is associated with caspase activation, we used western blotting to analyze the expressions of apoptosis-related caspases in NOB-treated BFTC cells, focusing on caspase-3 and caspase-9. Caspases and their associated kinases regulate cell survival and apoptosis. Inactivation of mitochondria results in cytochrome *C* release into the cytoplasm, followed by caspase-3 and caspase-9 activation to trigger apoptosis. Our results showed that the expressions of pro-caspase-3 and pro-caspase-9 were decreased after NOB treatment, while the expressions of activated cleaved-caspase-3, cleaved-caspase-9, and activated-PARP-1 were increased (Figure 3). In cells treated with NOB, the expression of caspase-8 was unchanged and the expressions of 14-3-3 protein decreased.

These results suggested that NOB-induced apoptosis is mediated via affecting the expressions of apoptosis-related proteins that belong to different pathways, including inactivation of the mitochondrial pathway and activation of the caspase-dependent pathway.

### 2.4. NOB-Induced Apoptosis Was Mediated by Endoplasmic Reticulum (ER) Stress

The aforementioned findings confirmed that NOB-induced apoptosis in BFTC cells is associated with mitochondria inactivation, involving increases in cytochrome *C*, Bad, and Bax proteins. Next, we explored the relationship between NOB and ER stress. When cells are under oxidative stress, unfolded proteins accumulate in the ER, causing ER stress. If the ER stress persists, unfolded protein accumulation initiates pathways that involve three ER signaling sensors, namely, PKR-like ER-associated kinase (PERK), inositol requiring enzyme-1α (IRE-1α), and activating transcription factor 6 (ATF6), which induce apoptosis. As shown in Figure 4, western blot analysis demonstrated that the expressions of protein disulfide isomerase (PDI), glucose-regulated protein GRP78, and calreticulin were increased with increasing concentrations of NOB, indicating that NOB treatment caused ER stress in BFTC cells. The purpose of upregulation of these proteins is to eliminate ER stress. The expressions of *p*-PERK, *p*-eIF2α, and downstream target proteins ATF4 and CHOP were all increased; the expression of pro-apoptotic protein ATF6 was also increased, while the expression of IRE-1α, which forms a complex with apoptosis signal-regulating kinase-1 (ASK-1) and activated JNK/c-Jun during apoptosis, was unchanged (Figure 4A). The results revealed that NOB-induced apoptosis was mediated by the PERK/eIF2α/ATF4/CHOP signaling pathway in the ER. In order to validate this statement, we added an eIF2α inhibitor, salubrinal, together with NOB for the treatment of BFTC cells, and examined whether apoptosis was induced in the cells. According to the outcome of an MTT assay, treatment with NOB together with salubrinal resulted in higher cell survival than treatment with NOB alone (Figure 4B), confirming that ER stress is involved in NOB-induced apoptosis in BFTC cells.

### 2.5. NOB Treatment Affected the PI3K/AKT/mTOR Signaling Pathway in BFTC Bladder Cancer Cells

Western blot analysis showed that the expressions of *p*-JNK, *p*-p38, and *p*-ERK remained the same in BFTC cells treated with NOB (Figure 5), suggesting that NOB-induced apoptosis is not mediated by the mitogen-activated protein kinase (MAPK) signaling pathway. Tian et al. [31] reported that the PI3K/AKT/mTOR signaling pathway is also involved in cell apoptosis, cell proliferation, cell differentiation, and cell survival. Our results indeed showed that the expressions of *p*-PI3K, *p*-AKT, and *p*-mTOR decreased with increased concentrations of NOB, but the expressions of PI3K, AKT, and mTOR remained unchanged (Figure 5), indicating that NOB-induced apoptosis occurs through inhibition of the PI3K/AKT/mTOR signaling pathway in BFTC cells.

## 3. Discussion

Screening of natural products to search for anti-tumor active compounds has become a popular area of research. Plants of the *Rutaceae* family have been found to contain polymethoxylated flavonoids with anti-cancer activity. NOB is a polymethoxylated flavonoid isolated from citrus peel that has been reported to inhibit tumor growth through inducing cell differentiation [32] and anti-cell proliferation [33,34], affecting the cell cycle and inducing apoptosis [22,35]. In the present study, the results demonstrated that NOB at concentrations of 60, 80, and 100 μM inhibited BFTC cell growth (Figure 1A), and the inhibition effect of NOB on the bladder cancer cell line was positively correlated with the concentration of NOB. Electrophoretic DNA analysis showed that NOB at a concentration of 60 μM caused apoptotic DNA fragmentation in BFTC cells (Figure 1B). In comparison with control cells, flow cytometry analysis revealed that in cells treated with 60 μM NOB, the early apoptotic rate was increased from 0.91% to 2.22%, and the late apoptotic rate was increased from 5.15% to 10.5%. These results suggested that NOB treatment caused late apoptotic cell death in BFTC cells (Figure 2).

To investigate the apoptosis pathway and the mechanism involved in NOB-induced apoptosis, we further studied the signaling pathways involved. Apoptosis occurs naturally in cells and is a useful strategy by which to develop cancer therapy [36]. Caspases are a family of cysteine proteases that play crucial roles in apoptosis. When death messages are received by death receptors embedded in the plasma membrane, inactive procaspase is cleaved to trigger downstream caspases, initiating a series of reactions. The induced downstream enzymes then destroy DNA, causing cell death. The caspase family of proteases can be divided into two groups—initiator caspases and executioner caspases. Initiator caspases include caspase-2, caspase-8, caspase-9, and caspase-10, while executioner caspases include caspase-3, caspase-6, and caspase-7. Active initiator caspases drive downstream executioner caspases to proteolytically cleave numerous regulatory and structural proteins, which initiates the cell death process [37]. Apoptosis is regulated principally by the Bcl-2 protein family. Pro-apoptotic Bcl-2 family members include multi-BH proteins (such as Bax and Bak), BH3-only proteins (such as Bad, Bim, Bid, and PUMA), and anti-apoptotic proteins (such as Bcl-2, Bcl-xl, and Mcl-1) [38]. Apoptosis can be triggered in cells by the intrinsic pathway or the extrinsic pathway [12,39]. Recent studies have shown that damage or stress to intracellular organelles, such as mitochondria and the ER, may induce the intrinsic pathway, causing cells to kill themselves. Studies have demonstrated that mitochondrial dysfunction plays a major role in apoptosis. Bax activation is important in ER-dependent apoptosis, as it leads to the formation of pores on the mitochondrial outer membrane and potentially causes mitochondrial inner transmembrane collapse [40]. In addition, study has also shown that an increase in the Bax/Bcl-2 ratio induces the release of cytochrome *C* and AIF from mitochondria, an increase in caspase-3, and elevation of cleaved PARP-1 protein, resulting in chromosome condensation and DNA fragmentation [15]. Therefore, once mitochondrial dysfunction has occurred, the intrinsic pathway is triggered to cause apoptosis, which destroys the balance between Bcl-2 and Bax, promoting cell survival by the pro-apoptotic pathway. As a result, expressions of pro-apoptotic proteins Bak, Bad, Bid, Bim, and PUMA are induced, while expressions of anti-apoptotic proteins Bxl-2, Bcl-xl, p-Bad, and Mcl-1 are decreased. This causes an increase in mitochondrial membrane permeability, cytochrome *C* release to the cytoplasm, and activation of caspase family proteins. In this pathway, caspase-3 and caspase-9 are especially known to play leading roles.

The western blotting results of our present study showed that the expressions of anti-apoptotic Bcl-2, Bcl-xl, Mcl-1, and *p*-Bad proteins were decreased in BFTC cells treated with increasing concentrations of NOB for 24 h, while the levels of pro-apoptotic Bad and Bax proteins, as well as cytochrome *C* release, were increased (Figure 3). The levels of active caspase-3 and caspase-9 were also increased, suggesting that NOB-induced apoptosis caused inhibition of DNA repair and reduced PARP-1 protein expression. Based on the above experimental results, NOB causes mitochondrial dysfunction and promotes apoptosis in BFTC cells.

The main functions of the ER are to regulate protein synthesis, protein folding, and calcium homeostasis [41]. If too many unfolded proteins are accumulated in the ER, the homeostasis of the ER will be destroyed; this will also cause intracellular organelle stress and lead to activation of a self-rescue program or triggering of apoptosis in cells. In the presence of stress, serval responses may occur in the ER, including ER-associated degradation (ERAD), the unfolded protein response (UPR), and apoptosis [42,43]. If stress in the ER continues and becomes severe, the UPR will increase, and apoptosis will be triggered [44]. Three ER signaling sensors—PKR-like ER-associated kinase (PERK), inositol requiring enzyme-1α (IRE-1α), and activating transcription factor 6 (ATF6)—are the key molecules that regulate the UPR. Increased accumulation of unfolded proteins in the ER will lead to ER chaperones, such as GRP78, releasing transmembrane proteins IRE-1α, ATF6 and PERK, which will result in ER dysfunction. The PERK signaling pathway can either promote cell survival via autophagy [45] or promote apoptosis through increasing ATF4/CHOP [46]. In cells under ER stress, the UPR will be triggered initially to activate a self-rescue program that can prevent cell death; however, if ER stress persists, pro-apoptotic responses will be initiated, which lead to cell death [47].

ER chaperones (GRP78, GRP94) and PDI are proteins that play major roles in the proper folding of proteins. Calreticulin is a molecular chaperone that performs the functions of quality control and synthesis of proteins [48,49]. Our western blot results demonstrated that the expressions of GRP78, calreticulin, and PDI were augmented under treatment with increasing concentrations of NOB, suggesting that these proteins were activated to reduce ER stress. When the UPR is inadequate to alleviate ER stress, the PERK, ATF6, and IRE-1α pathways may be activated to induce apoptosis in order to destroy ER stress-damaged cells [50]. Our results also showed that the expressions of *p*-PERK, *p*-eIF2α, and downstream ATF4 were increased, as was the expression of ATF6. The IRE-1α, *p*-JNK, *p*-c-Jun and level was unchanged. These findings suggested that NOB-induced apoptosis was mediated by the PERK/eIF2α/ATF4/CHOP signaling pathway in the ER. When salubrinal was used to treat BFTC cells together with NOB, inhibition of eIF2α by salubrinal significantly increased cell survival, confirming that NOB treatment initiated ER stress and subsequently induced apoptosis in BFTC cells.

MAPKs are a group of serine/threonine protein kinases that are involved in many cell responses, including cell proliferation, differentiation, and apoptosis. Three MAPKs—MAPK/ERK, SAPK/JNK and p38MAPK—are known to mediate multiple signal transduction pathways [51,52,53,54]. In our western blot analysis, NOB treatment did not alter the expressions of *p*-JNK, *p*-p38, or *p*-ERK in BFTC cells (Figure 5), indicating that the MAPK signaling pathway is not involved in NOB-induced cell death. Studies have demonstrated that the PI3K/AKT/mTOR signaling pathway is also involved in cell proliferation, differentiation, and apoptosis. Wang and coworkers [55] found that bardoxolone methyl caused mitochondrial inactivation and ER stress-mediated apoptosis though inhibition of the PI3K/AKT/mTOR pathway in human chronic myelogenous leukemia K562 cells. A study by Ke and Lou [56] showed that microRNA-10a, which can downregulate tyrosine kinase, suppressed breast cancer progression through regulating the PI3K/Akt/mTOR pathway. Our results demonstrated that the expressions of *p*-PI3K, *p*-AKT, and *p*-mTOR were inhibited under treatment with increasing concentrations of NOB (Figure 5), suggesting that NOB may inhibit BFTC cell proliferation and promote apoptosis through inhibiting the PI3K/AKT/mTOR signaling pathway.

In conclusion, the results of this study showed that NOB treatment caused late apoptotic cell death, and demonstrated that mitochondrial dysfunction and ER stress are involved in NOB-induced apoptosis. The apoptotic responses induced by NOB are mediated by regulating ER stress via the PERK/elF2α/ATF4/CHOP pathway, and the PI3K/AKT/mTOR pathway.

## 4. Materials and Methods

### 4.1. Chemicals and Antibody

Nobiletin, Z-VAD-FMK, Z-DEVD-FMK, dimethyl sulfoxide (DMSO), 3-(4,5-dimethyl-2-thiazolyl)-2,5-diphenyl-2-H-tetrazolium bromide (MTT) protease inhibitor cocktail, salubrinal, rabbit anti-human β-actin antibodies, and Dulbecco’s modified Eagle′s medium (DMEM) were purchased from Sigma (St. Louis, MO, USA). An annexin V-FITC/PI apoptosis detection kit was purchased from Pharmingen (San Diego, CA, USA). Antibodies against pro-caspase 9, cleaved caspase 9, caspase 8, pro-caspase3, cleaved caspase 3, Mcl-1, Bax, Bcl-xl, Bad, *p*-Bad, PARP-1, Bcl-2, 14-3-3, PDI, GRP78, calreticulin, cytochrome *C*, *p*-PERK, PERK, *p*-eIF2α, eIF2 α, ATF6-f, ATF4, IRE-1α, CHOP, *p*-JNK, *p*-c-jun, JNK, *p*-JNK, MAPKp38, *p*-MAPKp38, ERK, *p*-ERK, PI3K, *p*-PI3K, AKT, *p*-AKT, mTOR, *p*-mTOR, and goat anti-rabbit and horseradish peroxidase-conjugated immunoglobulin (Ig) G were obtained from Cell Signaling Technology (Danvers, MA, USA). Fetal bovine serum (FBS) was obtained from Biowest (Nuaillé, France).

### 4.2. Cell Culture

BFTC cells were purchased from the Taiwan Food Industry Research and Development Institute (Hsinchu, Taiwan). For maintaining the cells, we used Dulbecco’s modified Eagle’s medium (DMEM) medium with 10% fetal bovine serum (FBS), 100 μg/mL of streptomycin, and 100 units/mL of penicillin (Gibco BRL, Grand Island, NY, USA), and incubated at 37 °C in a humidified atmosphere of 5% CO_2_. Nobiletin dissolved in DMSO. Cells were treated with different concertation of nobiletin and harvested after 24 h of incubation. All experiments were performed three times to determine their reproducibility.

### 4.3. Cell Viability Assessment

BFTC cells were seeded in 96-well plates at a density of 1 × 10^5^/well for 24 h. The cells in each well were treated with nobiletin (at concentrations of 0, 20, 40, and 60 μM). After a 24 h incubation period, cell viability was assessed by MTT assay as follows: 50 μL of MTT (0.5 mg/mL) was added to the cells and incubated for 2 h. And then, removed the medium and added DMSO (200 μL) to dissolve the formazan. The optical density (OD) was monitored by ELISA reader (Bio-Rad, Hercules, CA, USA) at wavelength of 595 nm. All experiments were performed in at least triplicate.

### 4.4. Inhibitors Assessment

Investigation the effects of caspase-3 and caspase-9 on nobiletin-induced cell proliferation arrest. 1 × 10^5^ cells were seeded in a 24-well plate and pre-incubated for 2 h with Z-DEVD-FMK (caspase-3 inhibitor), Z-VAD-FMK (caspase-9 inhibitor), and salubrinal (eIF2α phosphorylation inhibitor) prior to nobiletin administration. Afterwards, the cell viability was determined by MTT assay.

### 4.5. Protein Lysate Preparation and Western Blot Analysis

After nobiletin treatment (0, 20, 40, and 60 μM) for 24 h, BFTC cells were lysed with a cell extraction buffer and protein concentrations were determined by the Bradford protein assay (Bio-Rad, Hercules, CA, USA). Total proteins (25 μg) were separated using 12.5% SDS-polyacrylamide gel electrophoresis (SDS-PAGE) gels, and transferred onto PVDF membranes, as described previously [33]. Mitochondrial and cytosolic cytochrome *C* was separated using a cytochrome *C* releasing apoptosis assay kit (Biovision, Milpitas, CA, USA). All membranes were incubated with primary antibody at 4 °C overnight. And then all membranes were followed by horseradish peroxidase-conjugated secondary antibodies (diluted 1:5000) for 2 h at 4 °C. The signals were detected through enhanced chemiluminescence detection kit (Pierce Biotechnology, Rockford, IL, USA).

### 4.6. Detection of Apoptotic Cells by Flowcytometry

Apoptotic cells within the BFTC cell lines induced by nobiletin (0, 20, 40, and 60 mM) after 24 h incubation were analyzed using an annexin V-FITC/PI apoptosis detection kit according to the manufacturer’s recommendations, as described previously [33]. Apoptosis processes induce from nobiletin were analyze by FACScalibur flow cytometer and Cell-Quest software (Becton-Dickinson, Mansfield, MA, USA).

### 4.7. Colony Formation Assay

BFTC cells were seeded in 24-well plates (2000 cells/well) and incubation for 24 h. The cells were treated with various concentrations (20, 40, and 60 μM) of nobiletin in 2 mL of serum complete media and incubation for 10 days, the colonies were then washed with PBS and fixed with methanol for 15 min and stained with 0.15% crystal violet. The colonies were counted and scanned by using a high-resolution scanner Scan Maker 9800XL (MiCROTEK, Hsinchu, Taiwain).

### 4.8. DNA Fragmentation Assay

Total DNA from BFTC cells treated with 20, 40, and 60 μM nobiletin for 24 h was extracted using the Suicide Track DNA ladder isolation kit. In total, 20 μg of DNA was analyzed on 2% agarose gels. The gels were detected by EtBr staining and the DNA band visualized under UV light (260 nm).

### 4.9. Wound Healing Assays

BFTC cells were seeded in 6-well plates (8 × 10^5^/well). After the cells reached confluence, a scratch made with a pipette tip in each well. Images of the control and experimental groups (0, 20, 40, and 60 μM nobiletin) were acquired at 0, 12, and 24 h after the treatments.

### 4.10. Statistical Analysis

Data for the cell viability assay, a flow cytometric analysis and colony formation assay were subjected by the ANOVA for analysis of variance, followed by Tukey’s test. Results with *p* < 0.05 were considered statistically significant.

## 5. Conclusions

Our results established that nobiletin possesses activity to induce apoptosis in BFTC bladder cancer cells are associated with mitochondrial dysfunction and endoplasmic reticulum stress. Nobiletin-induced apoptosis was also mediated by regulating endoplasmic reticulum stress via the PERK/elF2α/ATF4/CHOP pathway, and downregulating the PI3K/AKT/mTOR pathway (Figure 6). Our findings provided evidence that nobiletin has the potential to be developed as new therapeutic agent for bladder cancer treatment.

## Figures and Tables

**Figure 1 molecules-24-02881-f001:**
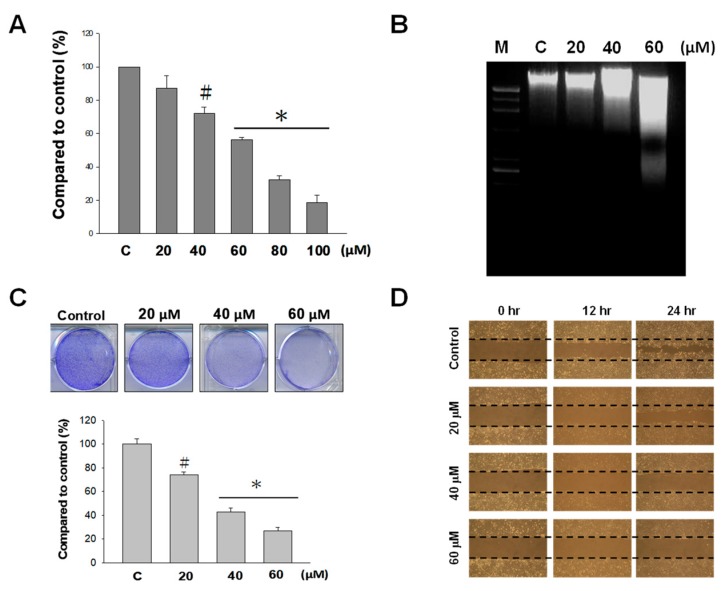
Effect of nobiletin (NOB) on cultures of BFTC bladder cancer cells. (**A**) BFTC cells were treated with NOB (20–100 μM) for 24 h, and the cytotoxic effect of NOB was analyzed by MTT assay. (**B**) DNA fragmentation caused by NOB treatment (20–60 μM) was detected via electrophoretic DNA analysis using agarose gel. (**C**) BFTC cells were treated with different concentrations of NOB (20–60 μM) for 10 days. After staining, the cell colony numbers were assessed by counting under a microscope. (**D**) After incubation with different concentrations of NOB (20–60 μM), a wound-healing assay was performed to analyze the inhibitory effects of NOB on BFTC cell proliferation. (#: *p* < 0.05; *: *p* < 0.01)

**Figure 2 molecules-24-02881-f002:**
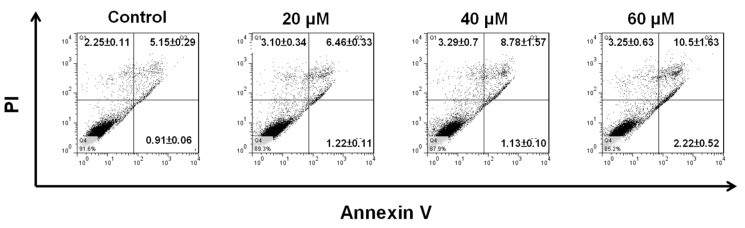
NOB induced apoptosis in BFTC bladder cancer cells. BFTC cells were treated with NOB at concentrations of 20, 40, and 60 μM for 24 h, following which they were double-stained with fluorescein isothiocyanate (FITC) Annexin-V and propidium iodide (PI). The percentages of early and late apoptotic cells were analyzed by flow cytometry.

**Figure 3 molecules-24-02881-f003:**
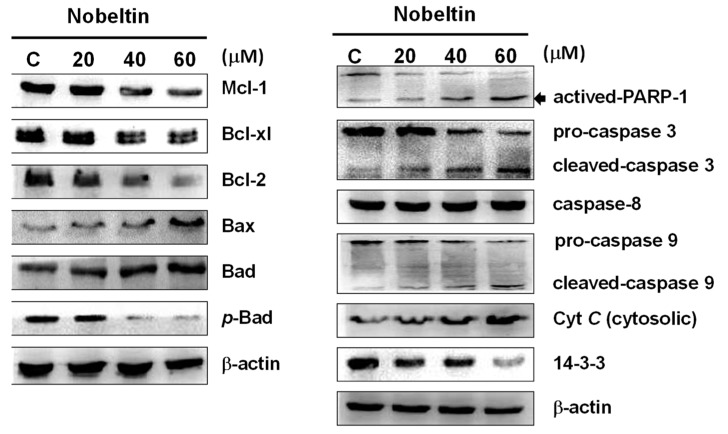
Western blot analysis of the expressions of proteins of the mitochondrial-mediated intrinsic pathway in NOB-treated BFTC bladder cancer cells. BFTC cells were treated with different concentrations of NOB (20–60 μM) for 24 h. The protein lysates were subjected to western blot analysis with antibodies against Mcl-1, Bcl-xl, Bcl-2, *p*-Bad, Bax, Bad, PARP-1, pro-caspase-3, cleaved-caspase-3, pro-caspase-9, cleaved-caspase-9, caspase-8, cytochrome *C*, and 14-3-3 protein. β-actin was used as the loading control.

**Figure 4 molecules-24-02881-f004:**
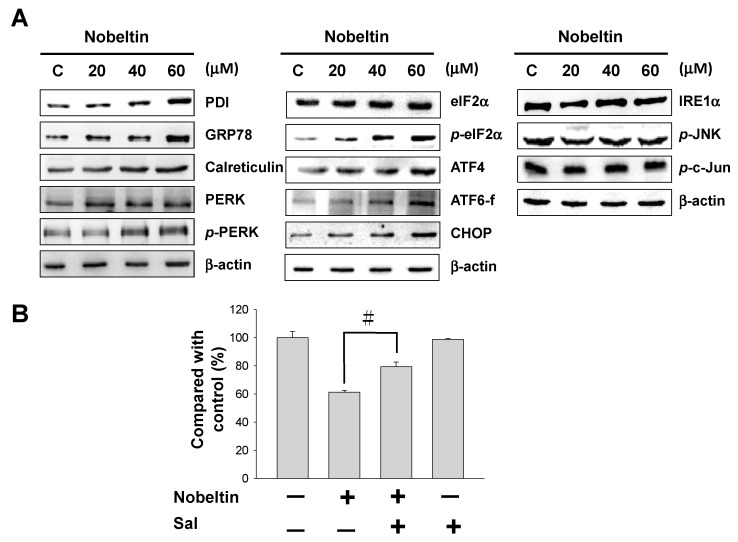
Western blot analysis of ER stress-related proteins in NOB-treated BFTC bladder cancer cells. (**A**) BFTC cells were treated with different concentrations of NOB (20–60 μM) for 24 h. The protein lysates were subjected to western blot analysis with antibodies against PDI, GRP78, calreticulin, PERK, *p*-PERK, eIF2α, *p*-eIF2α, IRE-1α, *p*-JNK, *p*-c-Jun, ATF4, ATF6-f, and CHOP proteins. An antibody to β-actin was used as the loading control. (**B**) Effect of salubrinal (an eIF2α inhibitor) on the survival of NOB-treated cells. Cells were treated with both NOB and 10 μM salubrinal, or with NOB alone, and cell survival was assessed using an MTT assay. Sal: salubrinal. (#: *p* < 0.05)

**Figure 5 molecules-24-02881-f005:**
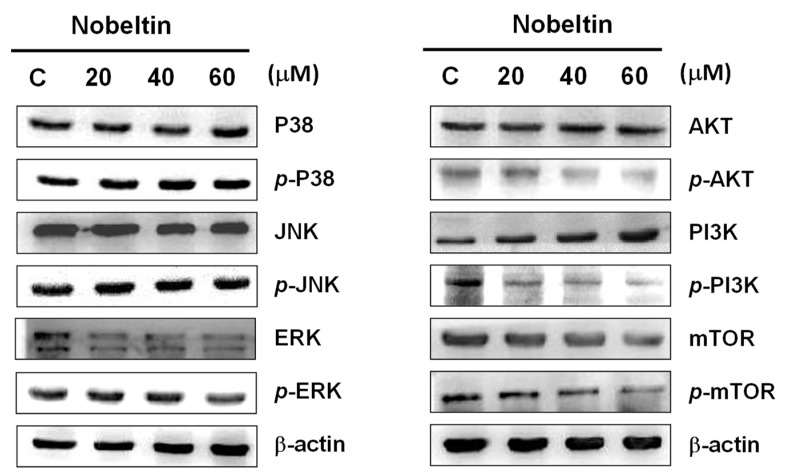
Western blot analysis of MAPKs and PI3K/AKT/mTOR signaling proteins in NOB-treated BFTC bladder cancer cells. BFTC cells were treatment with different concentrations of NOB (20–60 μM) for 24 h. The protein lysates were subjected to western blot analysis with antibodies against *p*-p38, p38, *p*-JNK, JNK, *p*-ERK, ERK, *p*-AKT, AKT, *p*-PI3K, PI3K, *p*-mTOR, and mTOR. An antibody to β-actin used as the loading control.

**Figure 6 molecules-24-02881-f006:**
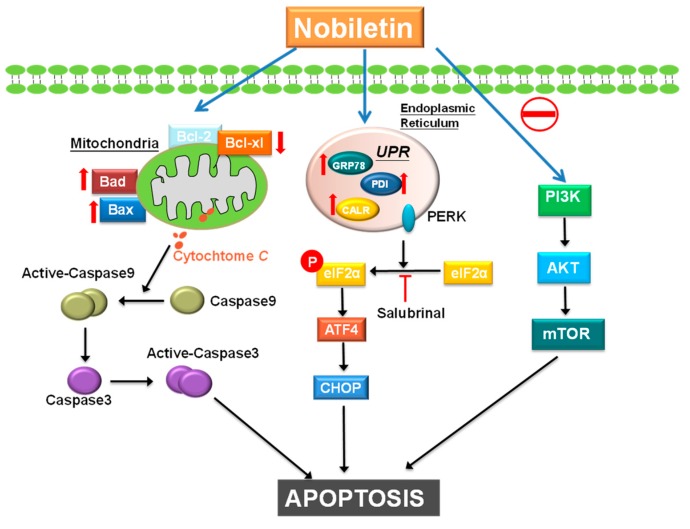
Nobiletin-induced apoptotic pathway in bladder cancer cells. The anti-cancer effect of nobiletin is mediated by the induction of mitochondria dysfunction and ER stress signaling pathways also involves downregulation of the PI3K/AKT/mTOR pathway.

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
