# Peer review of "Involvement of Mitochondrial Dysfunction, Endoplasmic Reticulum Stress, and the PI3K/AKT/mTOR Pathway in Nobiletin-Induced Apoptosis of Human Bladder Cancer Cells"

_molecules, 2019, doi:10.3390/molecules24162881_

Round 1

Reviewer 1 Report

The article “Involvement of mitochondrial dysfunction, endoplasmic reticulum stress and the PI3K/AKT/mTOR pathway in nobiletin-induced apoptosis of human bladder cancer cells” is interesting but it must be subject to major and minor revisions.

Major revision

1.      The author cannot suggest a mechanism of NOB on bladder cancer cells without performing at least cytotoxic assay on a normal cell line.

2.      Student’s test is not suitable to statistical analysis. You should use for example the ANOVA for analysis of variance, followed by Tukey’s test.  

Minor revisions

1.      In the section “Materials and Methods” it is not indicated the vehicle in which the NOB is dissolved.

2.      The English language needs to be improved.

3.      If it is possible the bibliography should be integrated with some more recent articles.

Author Response

Reviewer 1

Major revision

The author cannot suggest a mechanism of NOB on bladder cancer cells without performing at least cytotoxic assay on a normal cell line.

Respond: Thanks for reviewer’s suggestion. The concentrations of NOB in this study have been examined in normal skin cell lines (HaCaT cells). The results exhibite that the toxicity of Nobiletin. is lower in normal cells than in bladder cancer cells obviously.

Student’s test is not suitable to statistical analysis. You should use for example the ANOVA for analysis of variance, followed by Tukey’s test.  

Responds: Thanks for reviewer’s suggestion. We have modified the correct description in 4.10. Statistical Analysis

Minor revisions

In the section “Materials and Methods” it is not indicated the vehicle in which the NOB is dissolved.

Respond: Thanks for reviewer’s suggestion. We have modified mistake and expand description in line 348. Nobiletin dissolved in DMSO.

The English language needs to be improved.

Respond: Thanks for reviewer’s suggestion. This manucript has been improved by a native speaker.

If it is possible the bibliography should be integrated with some more recent articles.

Respond: Thanks for reviewer’s suggestion. We have supplement some reference in the manuscript. Reference 19-22.

Goh JXH, Tan LT, Goh JK, Chan KG, Pusparajah P, Lee LH, Goh BH. Nobiletin and Derivatives: Functional Compounds from Citrus Fruit Peel for Colon Cancer Chemoprevention. Cancers (Basel), 2019, 11(6): 867. Guney Eskiler G, Deveci AO, Bilir C, Kaleli S. Synergistic Effects of Nobiletin and Sorafenib Combination on Metastatic Prostate Cancer Cells. Nutr Cancer, 2019, doi: 10.1080/01635581.2019.1601237 Lin Z, Wu D, Huang L, Jiang C, Pan T, Kang X, Pan J. Nobiletin Inhibits IL-1β-Induced Inflammation in Chondrocytes via Suppression of NF-κB Signaling and Attenuates Osteoarthritis in Mice. Front Pharmacol, 2019, doi: 10.3389/fphar.2019.00570. Cirillo P, Conte S, Cimmino G, Pellegrino G, Ziviello F, Barra G, Sasso FC, Borgia F, De Palma R, Trimarco B. Nobiletin inhibits oxidized-LDL mediated expression of Tissue Factor in human endothelial cells through inhibition of NF-κB. Biochem Pharmacol, 2017, 128: 26-33.

Reviewer 2 Report

This is an interesting manuscript reporting the effects of a natural compound Nobuletin on human bladder cancer cells!the authors have shown that this novel compound induces apoptosis in these cancer cells and have dissected the pathways of both intrinsic and extrinsic mechanisms of apoptosis with evidence of mitochondrial damage,cytochrome C activation,Endoplasmic reticulum stress and involvement of PI3 kinase/Mtor pathway!this is a very well designed work and deserves immediate publication so more work could be done on this compound to study its pharmacokinetics,efficacy etc to be used as a successful anti cancer compound.

Author Response

Reviewer 2

This is an interesting manuscript reporting the effects of a natural compound Nobuletin on human bladder cancer cells!the authors have shown that this novel compound induces apoptosis in these cancer cells and have dissected the pathways of both intrinsic and extrinsic mechanisms of apoptosis with evidence of mitochondrial damage,cytochrome C activation,Endoplasmic reticulum stress and involvement of PI3 kinase/Mtor pathway!this is a very well designed work and deserves immediate publication so more work could be done on this compound to study its pharmacokinetics,efficacy etc to be used as a successful anti cancer compound.

Responds: Thanks for reviewer’s suggestion. We will continue the follow-up studies of pharmacokinetic studies on Nobiletin.

Reviewer 3 Report

In general, the manuscript is well-written and most of the data is clearly presented.  I have the following concerns and suggestions regarding this manuscript. 

1) All the data presented in the manuscript was obtained by utilizing ONE cancer cell line. To make the conclusions more convincing better to have a panel of cancer cells lines and non-transformed cells, as well (same origin). Especially, after taking into account the fact  that NOB affect mainly the cancer cells than non-transformed cells  (lines 77-78) 

2) lines 92-93 - the authors saying that they used the single concentration of NOB (60 micromolar) for the remaining experiments. However, all the further experiments were performed by using at least 3 concentrations of NOB (20, 40 and 60 micromolar).

3) the time period (12 and 24h) that the authors showed for cellular invasion assay is not correct and mainly reflects the proliferation capacities of cancer cells rather the invasion abilities. The authors do realize this point and arguing that NOB suppressed both processes in BFTCs - cell migration/proliferation (Fig 1D). However, a wound-healing assay is designed to examine the cellular invasion. Therefore, the data for 6 h should be presented on Fig 1D to reveal the inhibitory effects of NOB on invasion capacities of cancer cells.  

4) Flow cytometry data shown in Fig 2  requires an appropriate compensation between FL1 and 2 to remove the lines ("scratches") between the quandrants. Alternatively, the authors should try to work on FSC and SSC to remove this. Another concern about this data is a very weak proapoptotic effect of NOB and lack of dose-dependency in early apoptotic (e,g, PI-negative/AnnV-positive) cells. 

5)No data for the cleaved form of PARP is shown in Fig.3 This is a common apoptotic marker that is generally used with cl. caspase-3 to show the pro-apoptotic effects of the compounds.    

Author Response

Reviewer 3

In general, the manuscript is well-written and most of the data is clearly presented.  I have the following concerns and suggestions regarding this manuscript. 

All the data presented in the manuscript was obtained by utilizing ONE cancer cell line. To make the conclusions more convincing better to have a panel of cancer cells lines and non-transformed cells, as well (same origin). Especially, after taking into account the fact  that NOB affect mainly the cancer cells than non-transformed cells  (lines 77-78) 

Responds: Thank reviewer’s suggestion. We will continue study this topic with another bladder cancer in future.

lines 92-93 - the authors saying that they used the single concentration of NOB (60 micromolar) for the remaining experiments. However, all the further experiments were performed by using at least 3 concentrations of NOB (20, 40 and 60 micromolar).

Responds: Thank reviewer’s suggestion. We have corrected the mistakes.

the time period (12 and 24h) that the authors showed for cellular invasion assay is not correct and mainly reflects the proliferation capacities of cancer cells rather the invasion abilities. The authors do realize this point and arguing that NOB suppressed both processes in BFTCs - cell migration/proliferation (Fig 1D). However, a wound-healing assay is designed to examine the cellular invasion. Therefore, the data for 6 h should be presented on Fig 1D to reveal the inhibitory effects of NOB on invasion capacities of cancer cells.  

Responds: Thank reviewer’s suggestion. We have corrected several mistakes those are in line 102-105 and line 123.

Flow cytometry data shown in Fig 2  requires an appropriate compensation between FL1 and 2 to remove the lines ("scratches") between the quandrants. Alternatively, the authors should try to work on FSC and SSC to remove this. Another concern about this data is a very weak proapoptotic effect of NOB and lack of dose-dependency in early apoptotic (e,g, PI-negative/AnnV-positive) cells. 

Responds: Thank reviewer’s suggestion. We have modified the results. The results displayed that NOB did not cause early apoptosis, but it induced late apoptosis in BFTC cells. We remade new figure 2 in the manuscript.

5)No data for the cleaved form of PARP is shown in Fig.3 This is a common apoptotic marker that is generally used with cl. caspase-3 to show the pro-apoptotic effects of the compounds.    

Responds: Thank reviewer’s suggestion. We remade the western blotting analysis in figure 3.

Round 2

Reviewer 1 Report

The manuscript is suitable for pubblication in the present form.